# Antimicrobial Resistance Molecular Mechanisms of *Helicobacter pylori* in Jordanian Children: A Cross-Sectional Observational Study

**DOI:** 10.3390/antibiotics12030618

**Published:** 2023-03-20

**Authors:** Salma Burayzat, Mohammad Al-Tamimi, Mohammad Barqawi, Mustafa Sabri Massadi, Jumanah Abu-Raideh, Hadeel Albalawi, Ashraf I. Khasawneh, Nisreen Himsawi, Maha Barber

**Affiliations:** 1Department of Pediatrics and Neonatology, Faculty of Medicine, The Hashemite University, Zarqa 13133, Jordan; 2Department of Basic Medical Sciences, Faculty of Medicine, The Hashemite University, Zarqa 13133, Jordan; mohammad.altamimi@hu.edu.jo (M.A.-T.); salmajumanasami@gmail.com (J.A.-R.); hadeelkhaleel100@yahoo.com (H.A.); ashrafkh@hu.edu.jo (A.I.K.); nhimsawi@hu.edu.jo (N.H.); 3Faculty of Medicine, The Hashemite University, Zarqa 13133, Jordan; dr.mohammed1996@yahoo.com (M.B.); massadimustafa@gmail.com (M.S.M.); 4Department of Pediatrics, King Hussein Cancer Center, Amman 11941, Jordan; mbarbar@khcc.jo

**Keywords:** *Helicobacter pylori*, pediatrics, molecular, CLT-resistance, MTZ-resistance, RT-PCR, GyrA

## Abstract

Background: *H. pylori* antimicrobial resistance causes increasing treatment failure rates among *H. pylori* gastritis in children. This study investigates the molecular mechanisms of *H. pylori* antimicrobial resistance among Jordanian children. Methods: Demographic, clinical, and laboratory data were recorded for children referred to Prince Hamzah Hospital. Clarithromycin, Metronidazole, and Levofloxacin susceptibility were tested via E-test. Clarithromycin-related mutations were investigated using Real-Time (RT)-PCR and Levofloxacin resistance was analyzed with DNA sequencing of the *gyrA* gene. Results: 116 children were recruited, including 55.2% females and 55.2% in the age range of 10.1 to 14 years. A total of 82.7% were naïve to eradication therapy. *H. pylori* positivity was 93.9%, 89.6%, 61.7%, and 84.3% according to Rapid Urease Test, histology, culture, and RT-PCR, respectively. Resistance rates were 25.9% for Clarithromycin, 50% for Metronidazole, and 6.9% for Levofloxacin via E-test. A2142G or A2143G or a combination of both mutations concerning Clarithromycin resistance were documented in 26.1% of samples, while mutations in *gyrA* gen-related to Levofloxacin resistance were reported in 5.3% of samples. Antibiotic resistance was significantly affected by abdominal pain, anemia, hematemesis, and histological findings (*p* < 0.05). Conclusion: *H. pylori* resistance was documented for Metronidazole and Clarithromycin. RT-PCR for *H. pylori* identification and microbial resistance determination are valuable alternatives for cultures in determining antimicrobial susceptibility.

## 1. Introduction

*Helicobacter pylori* (*H. pylori*) is a spiral, Gram-negative, rod, curved, microaerophilic bacteria. It has multiple flagella, which play a crucial role in its motility and invasion mechanisms [1]. It was discovered in the human stomach in 1982 by Marshall and Warren, earning them the Nobel Prize for Medicine [1]. Typically, *H. pylori* infections are acquired during childhood and persist throughout life. It can cause chronic gastritis and gastroduodenal ulcers. Moreover, it is a significant risk factor for gastric cancer and mucosa-associated lymphoid tissue (MALT) lymphoma [2,3,4]. Its eradication effectively reduces the incidence of these malignant pathologies [2,5,6]. 

*H. pylori* is usually acquired in childhood, and the reinfection rate in children after its successful eradication is higher than in adults. However, compared to adults, children and adolescents with *H. pylori* infection rarely present with severe symptoms or develops serious pathologies [3,7]. 

The management of *H. pylori* gastritis can include various treatments such as using proton pump inhibitors (PPIs), antibiotics, bismuth, and probiotics. However, the number of antibiotics appropriate for *H. pylori* eradication in children is limited. The current recommended first-line *H. pylori* eradication regimens in children are mainly triple therapies consisting of a PPI plus two antibiotics chosen from Amoxicillin (AMX) or Clarithromycin (CLT or CLR) and Metronidazole (MTZ) for 14 days [7,8]. Levofloxacin (LVF or LEV) has restricted use in the pediatric age group due to safety concerns. On the other hand, Fluoroquinolones remain the drug of choice for specific indications, but their usage should be avoided in the presence of an alternative agent in children [9]. However, the eradication rate of these regimens has been less than 50%, especially among children [1]. On the other hand, many factors are responsible for eradication failure, starting with bacterial contamination, bacterial virulence, the CYP2C19 phenotype, patient compliance, and the most crucial factor: antibiotic resistance [10]. 

Given the limited number of antibiotics that are appropriate for *H. pylori* eradication in children, and the worldwide increase in antibiotic resistance, the recent European Society for Pediatric Gastroenterology Hepatology and Nutrition/North American Society for Pediatric Gastroenterology, Hepatology, and Nutrition (ESPGHAN/NASPGHAN) have recommended antimicrobial susceptibility testing to guide *H. pylori* eradication treatment in children [7,11]. However, due to the fastidious culture of *H. pylori* and the lack of an easy and cost-effective testing method, antimicrobial susceptibility testing for *H. pylori* is almost universally unavailable in medical centers [12,13,14,15,16]. Under such circumstances, profiling regional or population-specific antibiotic resistance patterns is crucial in guiding the development of effective empiric treatment regimens. As the *H. pylori* antibiotic resistance profiles among children and adolescents in Jordan are lacking, the aim of this study was to evaluate *H. pylori* strains isolated from this population for antibiotic resistance to CLR, MTZ, and LEV.

## 2. Results

### 2.1. Characteristics of Study Participants

Between January 2018 and June 2019, 116 children and adolescents were referred to pediatric gastroenterology clinics and underwent gastrointestinal endoscopy to rule out *H. pylori* infectious gastritis. A total of 52 (44.8%) males and 64 (55.2%) females were included in this study. Of these, 64 (55.2%) and 59 (50.9%) were between 10 and 14 years of age when conducting this study and at the time of diagnosis, respectively. A total of 90 (77.6%) children presented with chronic abdominal pain where vomiting was the presenting symptom in 20 (17.2%) children and refractory anemia was the presenting symptom in 9 (7.8%) children (Table 1).

In total, 106 (89.7%) children in the study population did not have a family history of *H. pylori* infection. Eighteen (15.5%) patients had taken at least one eradication course before undergoing endoscopy in our department, whereas ninety-six (82.7%) were naïve to eradication therapy. A total of 29 (25%) patients confirmed ever using an antibiotic: 10 (8.6%) patients used MTZ, 9 (7.7%) used a macrolide, 3 (2.5%) used MTZ and a macrolide, and 7 (6%) used other types of antibiotics (Table 1).

All of our patients underwent endoscopy, and RUT was positive in 109 (93.9%). It was negative in 4 (3.4%) patients, and it was not conducted in 3 (2.7%) patients. Histological examination of biopsies revealed severe gastritis in around 28 (24.1%) patients, while moderate gastritis was the predominant finding in 52 (44.8%) patients. *H. pylori* was found in 104 patients’ biopsies (89.6%) (Table 1). 

### 2.2. H. pylori Management and Follow Up

Of all patients, 103 (88.9%) had AMX, CLT, and PPI as their first eradication therapy. In comparison, 3 (2.6%) had AMX, CLT, and MTZ as the antibiotic part of their eradication therapy (Table 2). 

Forty-six (39.6%) patients were followed for one to six months. Seventy-eight (67.2%) were compliant with their treatment. We assessed the success of the first eradication therapy in 52 (44.8%) patients by evaluating symptoms and testing for *H. pylori* antigen in their stool. A total of 25 (21.6%) patients had positive *H. pylori* antigen in their stool after the first eradication therapy, whereas 27 (23.3%) patients had negative results. Of the total study population, 34 (29.3%) lost follow-up after the first eradication therapy (Table 2).

Twenty-six (22.4%) children of those who followed up needed second eradication therapy; eleven (9.5%) were given AMX, CLT, and MTZ. LVF and AMX were given as a second eradication therapy in 4 (3.4%) patients. Six (5.2%) patients were assessed for the success of the second eradication therapy; stool antigen was done in three (2.6%) of them and was positive in two (1.7%) patients (Table 2).

### 2.3. H. pylori Antimicrobial Resistance

There were 71 (61.2%) positive *H. pylori* cultures; 30 (42.3%) of them were resistant to CLT, 58 (81.7%) were resistant to MTZ, and 8 (11.3%) were resistant to LVF. Resistance was assessed using E-test MIC data. On the other hand, there were 38 (53.5%) cultures with combined resistance. A total of 26 (68.4%) cultures with combined resistance were resistant to CLT and MTZ, 7 (18.4%) were resistant to MTZ and LEV, 3 (7.9%) positive cultures were resistant to LEV and CLT, and 2 (5.3%) cultures were resistant to the 3 antibiotics studied: CLT, MTZ, and LEV (Table 3). 

DNA was extracted from 66 (92.9%) positive *H. pylori* culture plates and 50 (48.1%) positive *H. pylori* biopsies. An RT-PCR kit used specific urease gene primers (*ureA* and/or *ureB*) for *H. pylori* species identification relevant to its urease enzyme production and different primers for A2142G and A2143G in 23S rRNA operon for CLT resistance. Urease genes for *H. pylori* identifications using RT-PCR were documented in 97/115 (84.3%) extracted DNA samples. A2142G and/or A2143G mutations related to CLA resistance were documented using RT-PCR in 30/115 (26.1%) specimens (Table 4). 

LEV resistance was confirmed in 6/114 (5.3%) samples using DNA sequencing of the quinolone resistance-determining region (QRDR) of the *gyrA* gene (Table 4). Various substitutions were observed in the QRDR region of the *gyrA* gene, including four mutations corresponding to Asp-91 and two mutations corresponding to Asn-87. We also found two resistant strains with no QRDR mutations (Table 5).

### 2.4. Factors Associated with Failure of First Eradication Therapy or Antibiotics Resistance

Weight loss and abdominal pain as presenting symptoms were significantly associated with the failure of the first eradication therapy (*p* values 0.028 and 0.037, respectively). The previous eradication trial without complete diagnostics was significantly associated with the failure of the first eradication course (*p* value 0.011). Resistance to CLT detected by RT-PCR with a positive mutation was significantly associated with the failure of the first eradication therapy (*p* value 0.01) (Table 6).

Children between 10 and 14 years of age had the highest CLT resistance rate compared to other age groups (56.7%). No significant difference was found in resistance rates to MTZ, LEV, and CLT among the three age groups. The percentage of females with resistant strains to the three antibiotics studied, CLT, MTZ, and LEV, was higher (56.7%, 56.9%, 62.5%, respectively) (Table 7). 

Antibiotic resistance in *H. pylori* was significantly correlated with specific presenting symptoms. Abdominal pain was significantly associated with MTZ resistance (*p* value 0.034). Refractory anemia was associated significantly with CLT and LVF resistance (*p* values 0.048 and 0.018, respectively). On the other hand, hematemesis was significantly associated with CLT, MTZ, and LEV resistance (*p* values 0.003, 0.007, and 0.007, respectively) (Table 7).

A total of 100% of the strains resistant to LVF were found in *H. pylori* treatment-naïve children. The 34 strains isolated from patients with prior *H. pylori* treatment were more likely to be resistant to CLR than the 53 strains isolated from *H. pylori* treatment-naïve patients (70.6 vs. 45.3%, *p* = 0.02). However, no significant difference was observed between patients with a previous history of antibiotic use and resistant rates to MTZ and LEV (Table 7).

Twenty-six children (44.8%) had combined resistance to both CLT and MTZ, while three (37.5%) had combined resistance to CLT and LEV. A total of 26 patients (86.7% of patients with CLT resistance) and 7 patients (87.5% of patients with LVF resistance) had resistance to MTZ. The severity of histological findings in biopsies was significantly associated with resistance in all three antibiotics: CLT, MTZ, and LEV (*p* values 0.008, 0.005, and 0.01, respectively) (Table 7).

## 3. Discussion

The success of eradication therapy depends mainly on the antibacterial resistance pattern in a specific investigated population. Eradication failure can increase the risk of *H. pylori* resistance as well as burden patients with unnecessary extra procedures and increase healthcare utilization [17]. Primary CLT resistance has increased above the recommended levels (15%) for use as a first-line anti-*H. pylori* agent, demanding the use of regional antibiotic resistance data to govern its management [6,18]. 

This is the first longitudinal prospective study on *H. pylori* in Jordanian children. This fastidious microorganism was cultured, and its antimicrobial susceptibility was studied using phenotypic and molecular methods aiming at adding to the regional data on antimicrobial resistance in *H. pylori* to aid in the management of *H. pylori* gastritis in children.

*H. pylori* gastritis is the most common cause of gastritis and peptic ulcer disease worldwide. The infection is prevalent and increases with age. About 50% of the world’s population is estimated to have *H. pylori* infection [19,20]. Due to the variety of risk factors present in developing countries, infection with multiple *H. pylori* genotypes is highly prevalent in the Middle East and North Africa (MENA) region. The prevalence of *H. pylori* infection among the countries of the MENA region varies widely, ranging from 7 to 50% in young children and going up to 36.8–94% in adults [21], 86% in Saudi Arabia [22], and 70–82% in Jordan [23,24]. Limited data are available for the prevalence rate among pediatrics. *H. pylori* in gastric mucosa among pediatric patients who had endoscopy at Prince Hamzah Hospital was about 90% in this study. After reviewing reports from the last 10 years, it has been noticed that this percentage is consistent with data from many European countries, including France [25], Italy [26,27], Spain [28,29], and Portugal [30], and concurrent with the multicenter study on *H. pylori* primary resistance in Europe from 2013 [17]. 

This study shows a significant correlation between the presence of histological gastritis and *H. pylori* infection, especially for the moderate histological gastritis group, in agreement with previous studies [23,31]. Abdominal pain and weight loss were associated [12,14,32] with the failure of the first eradication therapy. Meanwhile, previous eradication therapy and CLT resistance significantly increased the risk of first-line therapy failure, which is consistent with the literature [33,34]. In this study, the *H. pylori* positivity rate was 93.9%, 89.6%, 84.3%, and 61.7% according to RUT, histology, RT-PCR, and culture, respectively. Other studies have shown similarly low rates for *H. pylori* culture due to its fastidious and demanding nature [12,13,14]. 

Among children, many global studies reported an increased incidence of primary antibiotic-resistant *H. pylori* strains [35]. The overall resistance rates in one study were 55.2% for CLT, 71.3% for MTZ, 60.9% for Rifampicin (RIF), and 18.4% for LVF [36].

The antimicrobial susceptibility test results of all *H. pylori* strains shown in Table 3 report a resistance rate in MTZ around 50% and 26% and 7% in CLT and LEV, respectively. Merei et al. showed resistance to MTZ in 32/46 isolates (69.5%), to CLT in 10 (21.7%), and to LEV in 3 (6.9%) [37]. Kalach et al. reported antimicrobial resistance in Iranian children as follows: MTZ, 62%; CLA, 22%; AMO, 16%; and LEV, 5.3% [38]. In Poland, the primary resistance of *H. pylori* indicates a persistently high level of CLT and MTZ resistance in both children and adults [39]. Meanwhile, LEV resistance increased over the last decade from 1.9% to 9.1% in pediatric patients and from 11.7% to 18.4% in adults [26]. This is in line with reports from another research center in Poland, indicating an increasing resistance of *H. pylori* strains to LEV [40]. Amoxicillin use is common among first-line therapy studied and evaluated by many investigations in adult and pediatric populations, and its resistance remains low. On the other hand, even though Levofloxacin has limited use in the pediatric age group due to safety concerns, it has specific indications as a “last resort” antibiotic after failures of other therapy lines. Multiple recent studies on antibiotic resistance in H. *pylori* gastritis in children have shown emerging levofloxacin resistance, which warrants investigation and more profound research. Recommendations regarding using levofloxacin in children have varied widely. Children resistant to levofloxacin will likely carry resistance that will mediate therapy failure in adulthood. Levofloxacin use in adults is becoming more common due to resistance to fist-line therapy. Many other studies conducted in children have also evaluated *H. pylori’s* resistance to Levofloxacin [34,37].

A2143G and/or A2142G mutation of 23S rRNA are the most common mediators of CLT resistance in *H. pylori* [37,39,40,41]. Similar findings were observed in this study. 

Specific mutations in the genes encoding DNA gyrase and/or topoisomerase IV cause fluoroquinolone resistance [42]. Mutations in the DNA gyrase gene have been assumed to be the origin of fluoroquinolone resistance in *H. pylori* caused mainly by point mutations in the QRDR of the *gyrA/B* gene [43]. There have been reports of *gyrA* mutations at Asn-87 and Asp-91 in the past [44,45]. We report mutations at Asp-91 in four resistant strains (50%) and mutations at Asn-87 in two resistant strains (25.0%). However, neither strain had both Asp-91 and Asn-87 mutations simultaneously. Mutations at Asp-91 were more frequent than at Asn-87. These findings are in accordance with earlier reports from Hong Kong and Vietnam, contrary to reports from Japan and China, indicating a correlation between geographical differences [45,46,47,48]. In this study, two resistant strains had no mutations in the QRDR of the *gyrA* gene but exhibited elevated MICs of fluoroquinolone antibiotics. This could be due to other mutations in the non-QRDRs of the *gyrA* or the less frequent *gyrB* gene, or other mechanisms, such as multidrug efflux systems [48,49].

## 4. Materials and Methods

### 4.1. Subjects

Children between one and fourteen years of age were referred to the pediatric gastrointestinal clinic at Prince Hamzah Hospital (PHH) from January 2018 to June 2019. Their presenting symptoms or laboratory findings suggested either *H. pylori* gastritis or peptic ulcer disease. Subjects were included in the study if they underwent Esophagogastroduodenoscopy (EGD), and a Rapid Urease Test (RUT) was performed during EGD and biopsies for histopathology and culture were taken. Children were excluded if they were older than 15 years at the time of EGD, had a severe systemic illness, were on PPI or antibiotics in the last month, or had previous gastric surgery.

### 4.2. H. pylori Diagnosis and Management

During EGD, six antral and two fundal gastric biopsies were obtained. The first two biopsies (one forceps pass) were used for culture, the second two biopsies were used for RUT, and the last four biopsies were used for histology [15]. A RUT was performed using a commercial kit (Helicotec UT plus, Taiwan). A color change from yellow to pink observed up to 4 h after the end of EGD indicated a positive result at 37 °C. 

Based on ESPGHAN/NASPGHAN guidelines, pediatric gastroenterologists diagnosed *H. Pylori* gastritis when tissue culture was positive for *H. pylori* or *H. pylori* was present in gastric mucosa in combination with a positive RUT. 

A negative *H. pylori* status was confirmed when histology and culture were negative. The 2016 ESPGHAN/NASPGHAN guidelines were followed to determine medical management regimens and dosages [7,11]. The treating physicians investigated and confirmed compliance with therapy. 

### 4.3. H. pylori Identification and Antimicrobial Susceptibility Testing 

Gastric biopsies collected from the antra and corpora of the patients were placed in sterile vials containing Stuart’s transport medium without charcoal (Biolab, Hungary) and transferred on ice to the Microbiology laboratory at the Faculty of Medicine at the Hashemite University. Biopsy tissues were homogenized, grounded using a tissue homogenizer (TissueLyser LT, Qiagen, Hilden, Germany), and inoculated into Columbia blood agar base (Oxoid, UK). This base contained 10% sheep blood and *H. pylori* selective supplement (Oxoid, UK) to inhibit the growth of other contaminant microorganisms. Plates were incubated at 37 °C and 5% CO_2_ in a CO_2_ incubator (Heracell™ 150i CO_2_, Waltham, MA, USA) for two weeks. Colonies displaying typical *H. pylori* morphology on agar were confirmed via Gram-staining, positive urease, catalase, and oxidase tests. The original biopsy samples and those that yielded positive *H. pylori* growth were stored in broth with 20% glycerol at –80 °C for further analysis. 

Two MacFarland’s standards of bacterial suspension were prepared and spread on solid agar media. Minimal Inhibition Concentration (MIC) was determined after incubation of plates for 5–7 days under proper conditions. *H. pylori* strains were tested for susceptibility to CLR, MTZ, and LEV using the Epsilometer test (E-test) strips (bioMérieux, Marcy-l’Étoile, France). According to the European Committee on Antimicrobial Susceptibility Testing [EUCAST], *H. pylori* resistance to CLR, MTZ, and LEV was defined as the MIC > 0.5 mg/L, >8 mg/L, and >1 mg/L, respectively [16].

### 4.4. Molecular Analysis of H. pylori and Antibiotics Resistance Genes

*H. pylori* genomic DNA was extracted and purified from biopsy samples or pure *H. pylori* colonies using a DNeasy Blood & Tissue extraction kit according to manufacturer instructions (Qiagen, Germany). Specific identification of *H. pylori* and detection of CLR resistance were performed using a Real-Time PCR (RT-PCR) detection Kit according to manufacturer instructions (VIASURE *H. pylori* + CLT resistance, Certest Biotec, Zaragoza, Spain). The kit uses specific urease genes (*ureA* and *ureB*) for *H. pylori* identifications and uses two major mutations, A2142G and A2143G, in 23S rRNA operon for CLT resistance. 

To determine the *gyrA* mutations, we amplified and sequenced the quinolone resistance-determining region (QRDR) of the *gyrA* gene. Primers used were *gyrA*F (5′-TTTRGCTTATTCMATGAGCGT-3′) and *gyrA*R (5′-GCAGACGGCTTGGTARAATA-3′). PCR was performed in a 50 μL reaction volume containing 2 μL of the template DNA and 2.5 U of OneTaq DNA Polymerase (New England Biolabs, Hitchin, UK). Thermocycling conditions were 94 °C for 5 min, followed by 35 cycles of 94 °C for 30 s, 53 °C for 30 s, and 72 °C for 30 s, with a final extension step of 72 °C for 10 min. 

The reaction products were checked and visualized by running 5 μL of the reaction mixture on 1% agarose gels. Sequencing of the amplified DNA was performed by the sequencing division at Princess Haya Biotechnology Center, University of Science and Technology, Irbid, Jordan. The sequences were then compared with the published sequence of the *H. pylori gyrA* gene (GenBank accession number L29481).

Informed consent was obtained from the patients’ parents or guardians. The IRB committee at The Hashemite University and the Jordan Ministry of Health/PHH approved this study. 

### 4.5. Statistical Analysis

Data was recorded and coded in Microsoft Excel 365 and then exported to Statistical Package for the Social Sciences (SPSS) version 25 for further analysis (IBM, Armonk, NY, USA, 2017). Categorical variables were presented as frequencies (numbers) and percentages (%). Continuous variables were further stratified into categories and presented as frequencies and percentages. Factors associated with failure of first eradication therapy and the presence of antibiotic resistance were analyzed using Chi-squared or Fisher’s exact test as appropriate, and *p* values less than 0.05 were considered significant.

## 5. Conclusions

This study provided important updates regarding *H. pylori* prevalence, antibiotic resistances rates, and significant risk factors for eradication treatment failure among Jordanian children. Drug-resistant *H. pylori* rates are increasing at an alarming rate that should be considered by treating physicians when planning eradication therapy. *H. pylori* resistance was documented for commonly used first-line eradication therapy associated with therapy failure and the required consideration of other treatment options. While *H. pylori* culture and antibiotic susceptibility are helpful tools in guiding treatment, technical difficulties and low success rate compromise their routine use to guide treatment. RT-PCR for *H. pylori* identification and microbial resistance determination are valuable alternatives. A regional antibiotic resistance profile for the population will help develop effective local eradication regimens.

## Figures and Tables

**Table 1 antibiotics-12-00618-t001:** Characteristics of patients.

Category	Variable		Number	Percentage (%)
**Demographics**	Age (years)	0–5	9	7.8
		5.1–10	43	37.1
		10.1–14	64	55.2
	Gender	Male	52	44.8
		Female	64	55.2
**Symptoms**	Abdominal pain		90	77.6
	Vomiting		20	17.2
	Diarrhea		13	11.2
	Weight loss		9	7.8
	Anemia		9	7.8
	Hematemesis		9	7.8
	Heartburn		7	6
	Flatulence		3	2.6
**Family history of *H. pylori***	Present not documented		1	0.9
	Present and treated without endoscopy		1	0.9
	Present and treated with endoscopy		5	4.3
	None		104	89.7
	N/A		5	4.3
**Previous eradication**	Yes		18	15.5
	No		96	82.7
**Antibiotics use**	Yes		29	25.0
	MTZ		10	8.6
	Macrolide		9	7.7
	Macrolide + MTZ		3	2.5
	Other		7	6
**Endoscopy**	Yes		116	100
	Macroscopic involvement of Stomach		114	98.3
	Macroscopic involvement of Duodenum		25	21.6
**Rapid Urease Test**	Positive		109	93.9
	Negative		4	3.4
	N/A		3	2.5
**Histology**	Normal		1	0.9
	Mild		29	25.0
	Moderate		52	44.8
	Severe		28	24.1
	N/A		6	5.2
** *H. pylori* ** **in Histology**	Yes		104	89.6
	No		9	7.8
	N/A		3	2.6
** *H. pylori* ** **culture**	Positive		71	61.2
	Negative		37	31.9
	N/A		8	6.9

N/A: Not available, MTZ: Metronidazole.

**Table 2 antibiotics-12-00618-t002:** *H. pylori* management and follow up.

Category	Variable	Number	Percentage (%)
**First eradication therapy**	PPI + AMX + CLT	103	88.9
	PPI + AMX + MTZ	1	0.9
	PPI + MTZ + CLT	2	1.7
	PPI + AMX + CLT + MTZ	3	2.6
	N/A	7	6.0
**Duration of follow-up (months)**	˂1	30	25.9
	1–6	46	39.6
	7–12	6	5.2
	The patient did not follow up	34	29.3
**Compliance with treatment**	Yes	78	67.2
	No	4	3.4
**Assessment of first eradication**	Resolution of symptoms	23	19.8
	*H. pylori* antigen in stool	7	6.0
	Symptoms resolution + stool *H. pylori* antigen	52	44.8
** *H. pylori* ** **antigen in stool after first eradication**	Positive	25	21.6
	Negative	27	23.3
	Not done	30	26.3
**Cause for the second course**	Recurrence of symptoms	9	7.8
	Positive stool *H. pylori* antigen	16	13.8
**Second eradication therapy**	Yes	26	22.4
	PPI + AMX + CLT	4	3.4
	PPI + AMX + MTZ	4	3.4
	PPI + MTZ + CLT	2	1.7
	PPI + AMX + CLT + MTZ	11	9.5
	LVF + AMX	4	3.4
**Assessment of the second eradication course**	Done and was negative for clearance	1	0.9
	Done and was positive for clearance	2	1.7
	Disappearance of symptoms	3	2.6
	N/A	4	3.4

PPI: Proton pump inhibitor, AMX: Amoxicillin, MTZ: Metronidazole, CLT: Clarithromycin, LVF: Levofloxacin. N/A: Not available.

**Table 3 antibiotics-12-00618-t003:** Culture and antimicrobial susceptibility using E-test.

Category	Variable	Number	Percentage (%)
**Culture**	Growth	71	61.2
	No growth	37	31.9
	Contamination	8	6.9
**Resistant to** **CLT**	Yes	30	42.3
	No	36	50.7
	N/A	13	18.3
**Resistant to MTZ**	Yes	58	81.7
	No	8	11.3
	N/A	13	18.3
**Resistant to LVF**	Yes	8	11.3
	No	58	81.7
	N/A	13	18.3
**Combined resistance**	CLT + MTZ	26	68.4
	MTZ + LVF	7	8.4
	CLT + LVF	3	7.9
	LVF + MTZ + CLT	2	5.3

AMX: Amoxicillin, MTZ: Metronidazole, CLT: Clarithromycin, LVF: Levofloxacin; N/A: Not available.

**Table 4 antibiotics-12-00618-t004:** DNA extraction and RT-PCR.

Category	Variable	Number	Percentage (%)
DNA extraction	From *H. pylori* culture	66	56.9
	From biopsy	50	43.1
*H. pylori* identification by RT-PCR	Positive	97	84.3
	Negative	18	15.7
Resistant to CLT by RT-PCR	Positive	30	26.1
	Negative	85	73.9
Resistant to LVF by DNA Sequencing	Positive	6	5.3
	Negative	108	94.7

RT: Real-time, PCR: Polymerase chain reaction.

**Table 5 antibiotics-12-00618-t005:** Amino acid changes in quinolone-resistant strains of *H. pylori*.

Strain	Mutation of *gyrA*
1R	Asp91Asn
2R	No QRDR mutation
3R	Asn87Lys
4R	Asp91Gly
5R	Asp91Asn
6R	Asp91Asn
7R	No QRDR mutation
8R	Asn87Lys

**Table 6 antibiotics-12-00618-t006:** Factors associated with failure of first eradication therapy.

Category	Variable	*p* Value
Demographics	Age (years)	0.635
	Age at diagnosis (years)	0.399
	Gender	0.263
	Address	0.319
Symptoms	Abdominal pain	**0.037 ***
	Vomiting	0.558
	Flatulence	0.406
	Heartburn	0.658
	Weight loss	**0.028 ***
	Anemia	0.682
	Hematemesis	0.682
	Diarrhea	0.489
Duration of symptoms (months)		0.148
Associated Medical diseases		0.635
Family history of *H. pylori*		0.466
Previous eradication		**0.011 ***
Antibiotics use		0.613
Resistant to CLT		0.061
Resistant to MTZ		0.747
Resistant to LVF		0.747
Resistant to CLT by RT-PCR with positive mutation		**0.010 ***

RT: Real-time; PCR: Polymerase-chain reaction; * Significant *p* value < 0.05.

**Table 7 antibiotics-12-00618-t007:** Factors associated with the presence of antibiotic resistance.

Variable	Category	CLR Resistance			MTZ Resistance			LEV Resistance		
		N	%	*p*	N	%	*p*	N	%	*p*
**Age**	0–5	3	10	0.978	5	8.6	0.99	0	0	0.99
	5.1–10	10	33.3		20	34.5		3	37.5	
	10.1–14	17	56.7		33	56.9		5	62.5	
**Gender**	Male	13	43.3	0.842	25	43.1	0.63	3	37.5	0.826
	Female	17	56.7		33	56.9		5	62.5	
**Presenting symptom**	Abdominal pain	23	76.7	0.669	47	81	**0.034 ***	7	87.5	0.57
	Vomiting	5	16.7	0.869	9	15.5	0.896	1	12.5	0.896
	Flatulence	2	6.7	0.37	3	5.2	0.606	0	0	0.606
	Heartburn	1	3.3	0.74	4	6.9	0.921	0	0	0.921
	Weight loss	3	10	0.86	6	10.3	0.759	0	0	0.759
	Anemia	0	0	**0.048 ***	2	3.4	0.08	1	12.5	**0.018 ***
	Hematemesis	1	3.3	**0.003 ***	5	8.6	**0.007 ***	0	0	**0.007 ***
	Diarrhea	4	13.3	1	7	12.1	1	1	12.5	1
**Previous eradication**	Yes	7	24.9	0.529	11	19.3	0.686	0	0	0.686
	No	22	75.9		46	80.7		8	100	
**Duration of symptom**	˂1	3	10	0.423	11	19.3	0.743	1	14.3	0.955
	1–6	10	33.3		14	24.6		1	14.3	
	7–12	7	23.3		12	21.1		3	42.9	
	˃12	9	30		17	29.8		2	28.6	
	Incidental finding	1	3.3		3	5.3		0	0	
**Family history of *H. pylori***	Yes	2	6.6	0.88	4	8.6	0.473	2	25	0.271
	No	28	93.3		53	91		6	75	
**Peptic ulcer disease**	Yes	0	0	0.316	1	1.7	0.326	0	0	0.326
	No	30	100		57	98.3		8	100	
**Antibiotics use**	Yes	7	25	0.122	15	27.3	0.102	2	25	0.127
	No	21	75		40	72.7		6	75	
**Resistant to CLR**	Yes	-	-	-	26	44.8	0	3	37.5	0
	No	-	-		32	55.2		5	62.5	
**Resistant to MTZ**	Yes	26	86.7	0	-	-	-	7	87.5	0
	No	4	13.3		-	-		1	12.5	
**Resistant to LVF**	Yes	3	10	0	7	12.1	0	-	-	-
	No	27	90		51	87.9		-	-	
**Macroscopic involvement Stomach**	Yes	26	96.7	0.80	58	100	0.76	8	100	1.0
	No	1	3.3		0	0		0	0	
**Macroscopic involvement Duodenum**	Yes	5	16.7	0.892	11	19	0.843	2	25	0.843
	No	25	83.3		47	81		6	75	
**Histology**	Normal	0	0	**0.008 ***	0	0	**0.005 ***	0	0	**0.010 ***
	Mild	6	20		11	19		0	0	
	Moderate	16	53.3		31	53.4		5	62.5	
	Severe	5	16.7		12	20.7		3	37.5	
	N/A	3	10		4	6.9		0	0	

MTZ: Metronidazole, CLT and CLR: Clarithromycin, LEV and LVF: Levofloxacin. * Significant *p* value < 0.05.

## Data Availability

Data is available from authors upon request.

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
