# Peer review of "Antimicrobial Resistance Molecular Mechanisms of Helicobacter pylori in Jordanian Children: A Cross-Sectional Observational Study"

_antibiotics, 2023, doi:10.3390/antibiotics12030618_

Round 1

Reviewer 1 Report

There are no adequate explication of statistical analysis, the method of conducting the statistical analysis should be described in more detail.

We are courious to know how come your study evaluated H.pylori strains isolated from pediatric population for antibiotic resistance to levofloxacin, which has restricted use in the pediatric age group, instead to evaluate resistance to amoxicillin, despite the fact that, in contrast to clarithromycin, metronidazole and levofloxacin, resistance to amoxicillin and tetracycline in adult population remained <10% in all WHO regions.

Author Response

Antimicrobial Resistance Molecular Mechanisms of Helicobacter pylori in Jordanian Children; A Cross-Sectional Observational Study

Authors: Salma Burayzat, Mohammad Al-Tamimi, Mohammad Barqawi, Mustafa Sabri Massadi Jumanah Abu-Raideh, Hadeel Albalawi, Ashraf I Khasawneh, Nisreen Himsawi, Maha Barber

Journal: Antibiotics

Manuscript ID: antibiotics-2260217

Reviewer 1:

Comments and Suggestions for Authors:

There are no adequate explication of statistical analysis, the method of conducting the statistical analysis should be described in more detail.

Response: Thank you. The statistical analysis section was added to the methodology section, page 9, as follows:

 "4.5 Statistical analysis

Data were recorded and coded in Microsoft Excel 365 and then exported to Statistical Package for the Social Sciences (SPSS) version 25 for further analysis (IBM, USA, 2017). Categorical variables were presented as frequencies (numbers) and percentages (%). Continuous variables were further stratified into categories and presented as frequencies and percentages. Factors associated with the failure of first eradication therapy and the presence of antibiotic resistance were analyzed using Chi-squared or Fisher’s exact test as appropriate. P values less than 0.05 were considered significant.”

We are curious to know how come your study evaluated H.pylori strains isolated from pediatric population for antibiotic resistance to levofloxacin, which has restricted use in the pediatric age group, instead to evaluate resistance to amoxicillin, despite the fact that, in contrast to clarithromycin, metronidazole and levofloxacin, resistance to amoxicillin and tetracycline in adult population remained <10% in all WHO regions.

Response: Amoxicillin use is common among first-line therapy studied and evaluated by many investigations in adults and pediatric populations, and its resistance remains low. On the other hand, even though Levofloxacin has limited use in the pediatric age group due to safety concerns, it has specific indications as a “last resort” antibiotic after failures of other therapy lines. Multiple recent studies on antibiotic resistance in H. pylori gastritis in children show emerging levofloxacin resistance, which warrants more investigation and more profound research. Recommendations regarding using levofloxacin in children varied widely. Children resistant to levofloxacin will likely carry resistance that would mediate therapy failure in adulthood. Levofloxacin use in adults is becoming more common due to resistance to fist lines therapy. Many other studies conducted in children have also evaluated H. pylori’s resistance to Levofloxacin [36, 39, 49, etc]. 

Line 141-146. Above paragraph was added to the Discussion.

Author Response

Reviewer 2:

Comments and Suggestions for Authors (was provided as a PDF file):

Line 39; “plus two antibiotics chosen from AMX (AML), CLT (CLR), and MTZ (MTZ) for 14 days[7,8. LVF (LEV) has restricted use in” please define antibiotic abbreviations here.

Response: antibiotics abbreviations were defined as requested in line 39.

Line 40: “On the other hand, Fluoroquinolones remain the drug of choice for specific

indications, and their usage should be avoided in the presence of an alternative

agent[9]” Confusing sentence; if drug of choice, “usage should be avoided…” here seems contradictory.

Response: The contradiction was removed, and the sentence was modified to read:

“Fluoroquinolones remain the drug of choice for specific indications, but their usage should be avoided in the presence of an alternative agent in children”

Line 43; perhaps “failure” should be used instead of “success”

Response: “failure” was used instead of “success” in line 43.

Introduction does a great job at justifying the study and making the case for the

importance

Response: thank you, no further action required.

Is observation of bacteria morphologically resembling H. pylori in histological section

sufficient to identify this species? Is this standard in the clinical field? Why not culture?

Response: yes, observation of H. pylori bacteria and its associated inflammatory changes in histological sections remain the gold standard for clinical diagnosis. Despite the benefits of H. pylori culture and antibiotics susceptibility testing, the fastidious nature, high rates of culture failure, and unavailability of this test hindered their clinical use.

The following text appears in the introduction, page 2, lines 48-49: “However, due to the fastidious culture of H. pylori and the lack of an easy and cost-effective testing method, antimicrobial susceptibility testing for H. pylori is almost universally unavailable in medical centers[12–14].”

The following text appears in the discussion section, page 8, lines 134-135: “Other studies have shown similarly low rates for H. pylori culture due to its fastidious and demanding nature[12–14].”

Also, stool antigen test done in a fraction of the post treatment children, but not all. This

information would be more valuable if done before eradication therapy.

Response: this is correct. Stool antigen was ordered for all patients. However, in a fraction of patients, stool antigen was not performed due to poor compliance of some patients.

Section 2.3 states 61.2% were culture positive. This seems like information that should have been in Table 1.

Response: thank you. This information was added to table 1, results section, page 3.

Results need to be much more clear regarding the use of ureAB in identification and

abx resistance. Reference to A2142G and A2143G mutations need clarity in terms of

gene sequenced. I know it is gyrA, but as written, could be interpreted as urease

mutations.

Response: thank you. The results section, page 5, lines 83-85 was clarified by adding the following text: “RT-PCR kit that uses specific urease genes primers (ureA and/or ureB) for H. pylori species identifications relevant to its urease enzyme production and different primers for A2142G and A2143G, in 23S rRNA operon for CLT resistance.”

The section on gyrA lines 86-87 was moved down and further clarified as follow: “LEV resistance was confirmed in 6 (5.2%) samples by DNA sequencing of quinolone resistance-determining region (QRDR) of gyrA gene (Table 4).”

Overall, this is a thorough and well-done study that presents massive amounts of

analyses that may be an important contribution to the field and I recommend publication

with only minor revisions.

Response: Thank you no further action required.

Salma Burayzat

Department of pediatrics and neonatology, Faculty of Medicine, The Hashemite University, Zarqa 13111, Jordan. Telephone: +962 (5) 3903333. Fax: +962 (5) 3826613. E-mail: salma.braizat@hu.edu.jo
